# The Cyclophilin Inhibitor Rencofilstat Decreases HCV-Induced Hepatocellular Carcinoma Independently of Its Antiviral Activity

**DOI:** 10.3390/v15102099

**Published:** 2023-10-17

**Authors:** Winston Stauffer, Michael Bobardt, Daren Ure, Robert Foster, Philippe Gallay

**Affiliations:** 1Department of Immunology and Microbiology, The Scripps Research Institute, La Jolla, CA 92037, USA; wstauffer@scripps.edu (W.S.); mbobardt@scripps.edu (M.B.); 2Hepion Pharmaceuticals Inc., Edison, NJ 08837, USA; dure@hepionpharma.com (D.U.); rfoster@hepionpharma.com (R.F.)

**Keywords:** HCV infection, hepatocellular carcinoma, cyclophilin inhibitor

## Abstract

There is an urgent need for the identification of new drugs that inhibit HCV-induced hepatocellular carcinoma (HCC). Our work demonstrates that cyclophilin inhibitors (CypIs) represent such new drugs. We demonstrate that the nonimmunosuppressive cyclosporine A (CsA) analog (CsAa) rencofilstat possesses dual therapeutic activities for the treatment of HCV infection and HCV-induced HCC. Specifically, we show that the HCV infection of humanized mice results in the progressive development of HCC. This is true for the four genotypes tested (1 to 4). Remarkably, we demonstrate that rencofilstat inhibits the development of HCV-induced HCC in mice even when added 16 weeks after infection when HCC is well established. Importantly, we show that rencofilstat drastically reduces HCC progression independently of its anti-HCV activity. Indeed, the CypI rencofilstat inhibits HCC, while other anti-HCV agents such as NS5A (NS5Ai) and NS5B (NS5Bi) fail to reduce HCC. In conclusion, this study shows for the first time that the CypI rencofilstat represents a potent therapeutic agent for the treatment of HCV-induced HCC.

## 1. Introduction

The name “cyclophilin” (Cyp) comes from the discovery of cyclophilin A (CypA) as a ligand of cyclosporine A (CsA) [1] and its peptidyl-prolyl isomerase (PPIase) activity [2]. Cyps belong to a family of enzymes that catalyze the *cis-trans* isomerization of proline peptide bonds, which are unique from other bonds due to their ability to switch between *cis* and *trans* conformations. The Cyps-mediated catalyzation of *cis-trans* proline peptide bonds occurs, at many orders, faster than uncatalyzed *cis-trans* proline peptide bonds [3,4]. Prolines shape the configuration of proteins due to the rigidity of the pyrrolidine ring of prolines. Cyps, by catalyzing the *cis-trans* isomerization of proline peptide bonds, regulate the structural conformation of proteins, their ligand-binding properties, and biological functions. Thus, Cyps participate in a broad range of activities, including (i) the initial folding of nascent peptides into proteins; (ii) the restraint of protein aggregation; (iii) intracellular protein trafficking and secretion; (iv) the amplification of the second messenger signaling; and (v) the regulation of protein–protein interactions [5,6,7,8]. The immunosuppressant CsA came into medical use in 1983 to prevent graft-versus-host disease [9]. The immunosuppressive activity of CsA originates from CsA binding to CypA and the formation of a ternary complex with calcineurin, which blocks the activation of the nuclear factor of activated T cells and its downstream signaling. CsA neutralizes the isomerase activity of CypA and other Cyp members such as CypB, CypC, and CypD with high affinities (Ki of ~15 nM) due to their highly conserved enzymatic pockets [10]. The chemical alteration of CsA resulted in the identification of CsA analogs (CsAas) devoid of immunosuppressive activity but with preserved CypA-binding activities [11,12,13]. Four Cyp inhibitors (CypIs) have been studied in humans—the CsA analogs (CsAas) NIM811, alisporivir (ALV)/Debio-025, SCY-635, and rencofilstat (CRV431). Rencofilstat potently inhibits all cyclophilin isoforms tested, namely cyclophilin A, B, D, and G [14]. Inhibitory constant or IC_50_ values ranged from 1 to 7 nM, which was up to 13 times more potent than the parent compound, CsA from which rencofilstat was derived. Other rencofilstat advantages over CsA as a nontransplant drug candidate include significantly diminished immunosuppressive activity, less drug transporter inhibition, and reduced cytotoxicity potential [14]. Oral dosing to mice and rats led to exposures expected to be in line with therapeutic blood concentrations, and a 5- to 15-fold accumulation of rencofilstat in the liver compared with whole-blood concentrations across a wide range of rencofilstat dosing levels [14]. More recently, rencofilstat was safe and well tolerated after 28 days in subjects with presumed F2/F3 NASH [15]. The presence of NASH did not alter its pharmacokinetics.

We and others showed that CsAas inhibit HCV in vitro [16,17,18,19,20,21,22,23,24,25,26,27,28,29,30,31,32,33]. ALV showed high anti-HCV efficacy in humans in phase I-III studies [34,35,36,37]. We reported that HCV fails to infect CypA knockdown (KD) cells, while it infects CypB-KD, CypD-KD, and parental cells [26], suggesting that HCV requires CypA to optimally replicate in human hepatocytes. Supporting this notion, the Ploss lab demonstrated that HCV fails to infect CypA knockout (KO) humanized mice [38]. We showed that the HCV infection of CypA-KD cells is restored after the reintroduction of wild-type CypA but not isomerase-deficient CypA (H126Q mutation in the enzymatic pocket of CypA) [26], further suggesting that HCV relies on the foldase activity of CypA to replicate in cells. We demonstrated that CsAas inhibit HCV infection by preventing NS5A-CypA interactions and preventing the formation of ER double-membrane vesicles (DMVs) required to shield the amplification of the viral RNA genome [28]. Thus, the requirement for CypA in HCV infection is well understood at a cellular level. Moreover, we demonstrated that rencofilstat inhibits HCV-induced HCC in humanized mice and that viral replication was totally suppressed after 4 days of rencofilstat treatment [39]. Altogether these data strongly suggest that rencofilstat inhibits HCV infection by sequentially neutralizing the PPIase activity of CypA, preventing CypA-NS5A interactions, leading to the suppression of CypA-NS5A-mediated DMV formation that normally protects the viral genome replication and viral particle formation.

Anti-HCV treatments have progressed significantly during the last 20 years. Current HCV treatments include highly successful combinations of pangenotypic direct-acting antivirals (DAAs) with short-period treatments (8–12 weeks), high sustained virological response (SVR) (>95%), and minimal side effects. However, the combination of specific viral genotypes (GTs) and disorder conditions diminishes SVR levels to DAA such as HCV genotype 3 (GT3) infection, cirrhosis, and DAA resistance associated with the selection of resistance-associated substitutions (RASs) present at baseline or are acquired during treatment [40]. An option to avoid the retreatment of HCV patients for viral resistance and GT3 infection would be to incorporate into current FDA-approved DAA regimens, antivirals with high barriers to resistance and with different antiviral mechanisms of action (MoA). The CypI rencofilstat possesses such unique antiviral activities.

HCC is the third major cause of cancer mortality, globally accounting for 800,000 deaths per year [41]. Moreover, HCC is one of the fastest-growing causes of cancer mortality in the United States [41]. Current treatment strategies for HCC include liver transplantation, segmentectomy, chemotherapy, and systemic drug therapy. However, tumor recurrence may occur after transplantation and segmentectomy, and HCC is minimally responsive to chemotherapy. Moreover, chemotherapy presents numerous toxic side effects [42,43]. Importantly, unexpectedly high rates of HCC recurrence occur after hepatic resection [44] and chemotherapy [45,46]. Therefore, the identification of new effective treatments for HCC is a crucial research interest. We recently demonstrated and reported that rencofilstat decreases liver fibrosis in a nonviral 6-week carbon tetrachloride model as well as in a mouse model of nonalcoholic steatohepatitis (NASH) [14]. Rencofilstat administration during a late, oncogenic stage of the NASH disease model results in a 50% reduction in the number and size of liver tumors [14]. These findings are consistent with rencofilstat targeting fibrosis and cancer via multiple Cyp-mediated mechanisms and re-emphasize the utmost importance of exploiting rencofilstat as a safe and effective drug candidate for the treatment of liver diseases.

Approximately, 71 million people worldwide are infected with HCV [47,48,49]. In HCV patients, liver damage varies, from negligible injuries to pronounced fibrosis, cirrhosis, or HCC. Worldwide deaths associated with HCV-induced complications were estimated to be ~600,000 in 2017 [20]. In this study, we investigated for the first time whether rencofilstat could inhibit HCC development induced by HCV infection rather than induced by chemicals or a high-fat diet. We found that a daily rencofilstat treatment of mice dramatically diminished HCV-induced HCC in humanized mice even when added months after infection, when viral replication was robust and HCC was established, suggesting that rencofilstat could be a novel therapeutic drug for the treatment of HCV-induced HCC in HCV patients infected even for several months.

## 2. Materials and Methods

### 2.1. Drugs and Antibodies

Rencofilstat (CRV431) was synthesized via the chemical modification of cyclosporin A with a purity > 95% determined using HPLC. The NS5B polymerase inhibitor (NS5Bi) sofosbuvir and the NS5A inhibitor velpatasvir (NS5Ai) were purchased from MedChemExpress. Antimurine MCP-1 (2D8 mAb), TMP-1 (10D1 mAb), GP3 (9C2 mAb), and GAPDH (ZG003 mAb) antibodies were purchased from Thermo Fischer Scientific (Waltham, MA, USA). The mouse alanine aminotransferase (ALT) ELISA kit (ab285263, E4324) and the mouse AST ELISA Kit (aspartate aminotransferase) (ab263882) were purchased from Abcam.

### 2.2. Animal Care

Animal housing: Mice were housed in individually ventilated cage (IVC) racks. HEPA-filtered air was provided into the solid bottom cages at a rate of 60 air changes per hour. Static mouse cages and IVCs were changed every week and every two weeks, respectively. To ensure a proper room environment, air conditioning, ventilation, and heating equipment were routinely tested. Animal rooms that contained a high/low thermo-hygrometer with its computerized controlled thermostat for humidity were monitored daily. Consistent with Guide recommendations, temperature and alarm points were set at ±4 °F by the Engineering Department. In addition to the automated building management system, animal facilities were equipped with an Edstrom Industries Watchdog environmental monitoring system, which registers humidity and temperature and sends alarms to Animal Resources personnel.

Diet: Teklad LM-485 autoclavable diet food was given ad libitum to mice in wire bar lids. Water: The vivarium was supplied with a reverse osmosis (R/O) water purification system and an automatic watering distribution system from Edstrom Industries that were monitored daily. Temperature, pH level, chlorine concentration, and conductivity were routinely checked. Automatic water delivery systems were daily timed for in-line flushing and were sanitized when needed by the DAR equipment technicians.

Acclimation period: Mice were acclimated for three days into their new housing environment.

Animal suffering: All efforts were made to minimize suffering. All surgical procedures were conducted under anesthesia using isoflurane (1–4%) with ketamine/xylazine ip (90 mg/kg and 10 mg/kg). Mice were checked every 15 min for respiratory and heart rates. As a postoperative analgesic, animals were given buprenorphine (0.05–2.5 mg/kg s.c.) for 6–12 h and flunixine meglumine (2.5 mg/kg s.c.) for 2 days post-engraftment. Mice were carefully monitored on the day of surgery and then daily during the study. To decrease the possibility of opportunistic bacterial colonization, acidified water supplemented with sulfamethoxazole (or sulfadiazine) with trimethoprim at a final concentration of 0.65–1.6 mg/mL was provided to mice. MUP-uPA-SCID/Beige mice were housed at TSRI in accordance with protocols approved by the TSRI Ethics Committee and the Institutional Animal Care and Use Committee (Protocol Number: 11–0015). This study was conducted in strict accordance with the recommendations in the Guide for the Care and Use of Laboratory Animals of the National Institutes of Health. Mice were sacrificed via cervical dislocation. A power calculation was used to determine ten mice per group for each treatment.

### 2.3. HCV Chimeric Mouse Study

MUP-uPA-SCID/Beige mice (gift from A. Kumar), which have the uPA gene driven by the major urinary protein promoter [50], can be engrafted with human hepatocytes until the age of 12 months. MUP-uPA-SCID/Beige mice (4 months old) were engrafted with 10^7^ human hepatocytes per mouse. We normally acquired ~300–500 × 10^6^ hepatocytes per donor from D. Geller as we reported previously [39]. After viability confirmation, fresh human hepatocytes were engrafted straightaway upon arrival within 16 h post-isolation. A skin incision (1 cm) was made in the upper left abdomen quadrant to inspect the spleen for the intrasplenic injection of human hepatocytes. Vetbond tissue adhesive (3M Animal Care Products, St. Paul, MN, USA) was used to close the incision. Human albumin (hAlb) blood levels were quantified using ELISA (Bethyl Laboratories, Montgomery, TX, USA) according to the manufacturer’s protocol to verify the degree of “humanization” of the animals. Mice expressing >300 μg/mL of hAlb were selected and randomized into groups (n = 10). Although the uPA transgene expression induces murine liver damage, engrafted human hepatocytes visualized via hAlb immunostaining rapidly restore the functionality of the humanized liver [50]. We confirmed that enhancing the number of engrafted human hepatocytes is critical to obtaining functional repopulated human hepatocytes by quantifying serum hAlb via ELISA a month after engraftment [50]. Humanized liver mice were then infected intravenously (i.v.) with serum from infected chimpanzees (100 infectious doses): HC-TN GT1a, HC-J6 GT2a, S52 GT3a, and ED43 GT4a (gift from Dr. Lanford). CRV431 was dissolved in PEG-300, while velpatasvir and sofosbuvir were dissolved in DMSO and subsequently in a 95% sterile saline solution. Drugs were administered once by oral gavage at 50 mg/kg, and blood was collected retro-orbitally at the indicated time points.

### 2.4. Quantification of HCV RNA by Real-Time Reverse Transcription PCR

HCV RNA in serum and livers was isolated using the acid guanidinium–phenol–chloroform method. Real-time reverse transcription PCR (RT-PCR) TaqMan chemistry was used to quantify HCV RNA as we reported previously [51].

### 2.5. HCC Analyses

Liver tumors from livers collected 30 weeks post-HCV infection were counted, and their diameters were quantified and cataloged according to their diameter as small (<0.5 cm), medium (0.5–1 cm), or large (>1 cm). Each liver was scored for tumor burden (0–7 scale) based on criteria that we described previously [14,52]. The cancerous status of each nodule was verified via qRT-PCR for tumor markers MCP-1, Timp-1, and glypican-3. RplP0 (Mus musculus ribosomal protein, large, P0) was used as the control. The forward primer for MCP-1 was 50-GCATCCACGTGTTGGCTCA-30, and the reverse primer was 50-CTCCAGCCTACTCATTGGGATCA-30. The forward primer for Timp-1 was 50-TGAGCCCTGCTCAGCAAAGA-30, and the reverse primer was 50-GAGGACCTGATCCGTCCACAA-30. The forward primer for glypican-3 was 50-CCAGATCATTG.

ACAAACTGAAGCA-30, and the reverse primer was 50-CGCAGTCTCCACTTTCAAGTCC-30. The forward primer for 36B4 was 50-TTCCAGGCTTTGGGCATCA-30, and the reverse primer was 50-ATGTTCAGCATGTTCAGCAGTGTG-30. To correct the variation in the amount of cDNA available for PCR in the different samples, gene expressions of the target sequence were normalized in relation to the expression of an endogenous control, 36B4 mRNA.

## 3. Results

### 3.1. HCV Infection Induces the Development of HCC in Humanized Liver Mice

The main goal of this study is to determine whether the CypI rencofilstat prevents HCV-induced HCC. We first examined whether HCV infection replication from genotypes 1 to 4 (GT1a-GT4a) in humanized mice elicits HCC. As we previously reported [39], HCV replication reaches maximal levels 3 weeks post-infection and remains stable for 30 weeks (Figure 1A). This is true for the four genotypes tested. The four genotypes induced a similar degree of HCC (Figure 1B–D) after 30 weeks.

### 3.2. HCV Infection Induces a Progressive HCC Development in Humanized Liver Mice

We next examined the kinetics of the HCV-induced development of HCC. HCV (GT1a)-induced HCC development started 12 weeks post-infection, as demonstrated by the occurrence of small cancerous nodules (Figure 2B), and expanded over time, as demonstrated by the higher numbers of small cancerous nodules by 16 weeks and larger cancerous nodules by 24 weeks (Figure 2C). We confirmed HCC by analyzing Western blot of the expression of cancer markers—MCP-1, TIMP-1, and GP3—in either isolated nodules from HCV-infected mice or from liver tissue from uninfected mice. As expected, we observed a high expression of cancer markers in nodules but not in the normal liver (Figure 2D), confirming the HCC status observed via qPCR. Similar GPDH levels indicate that similar amounts of liver tissue were analyzed (Figure 2D). These data further confirm our qPCR data indicating that the liver nodules were cancerous. This demonstrates that HCV infection induces progressive HCC development in humanized MUP-uPA-SCID/Beige mice. These findings are critical because they provide evidence of successive stages of HCV-induced HCC progression and therefore more precisely map when rencofilstat treatment can be initiated.

### 3.3. Rencofilstat Treatment Reduces HCV-Induced HCC

We next examined the effects of rencofilstat on HCV-induced HCC. GT1a HCV-induced HCC development was initiated as described above. Daily rencofilstat treatment initiated at week 0 completely prevented HCC development measured at week 30 (Figure 3A,B). This was expected since we have previously shown that rencofilstat inhibits HCV infection and therefore eliminates the driver of HCC development [39]. In other mice, daily rencofilstat administration was initiated when HCC was present in its early, nodular stage (week 12); mid-stage with medium-sized tumors (week 16); or large, late-stage tumors (week 24), as shown in Figure 2. Importantly, daily rencofilstat administration initiated at week 12 not only prevented the development of additional nodules but also appeared to regress existing small nodules, since no HCC was observed at week 30 (Figure 3A,B). More importantly, rencofilstat treatment initiated at week 16, when HCC was well established, significantly decreased tumor sizes at week 30 (Figure 3A,B). These data indicate that rencofilstat possesses anti-HCC activity, including some degree of regressive activity. Both cancer nodule numbers and sizes were impacted by rencofilstat treatment. Specifically, rencofilstat treatment initiated at week 0 or 12 totally prevented nodule formation. Rencofilstat treatment initiated at week 16 (nodule numbers ~2.0 and nodule size ~3.4) and 24 (nodule numbers ~3.5 and nodule size ~10.5) reduced both the nodule numbers and sizes. Note that the average nodule number in vehicle-treated mice is ~8.3 [14,15]. We also verified that rencofilstat treatment rapidly suppresses HCV replication (Figure 3C). This dual beneficial therapeutic effect—anti-HCV and anti-HCC—highlights rencofilstat as a promising anti-HCV-induced HCC therapeutic agent. We also analyzed the weight (Figure 3D) and AST/ALT levels (Figure 3E,F) in HCV-infected mice treated with rencofilstat over the entire 24 weeks of treatment. In accordance with rencofilstat treatment of NASH patients [15], our results further suggest that a rencofilstat dose of 50 mg/kg/day does not induce significant liver toxicity.

### 3.4. Rencofilstat’s Anti-HCC Activity Is Partly Independent of Its Anti-HCV Activity

To investigate whether some of rencofilstat’s anti-HCC effects occur independent of its anti-HCV activity, we compared rencofilstat to other anti-HCV agents in the model. Specifically, we investigated the anti-HCC effect of the NS5Bi sofosbuvir and the NS5Ai inhibitor velpatasvir. As expected, rencofilstat, sofosbuvir, and velpatasvir daily treatments starting at 16 weeks post-infection, when HCV replication was maximal and when HCV-induced HCC was well established, totally suppressed HCV replication at week 30 (Figure 4A, last column). Remarkably, rencofilstat, but not sofosbuvir or velpatasvir, also significantly decreased the number of tumor nodules at week 30, by approximately 80% (Figure 4B). The tumors that remained in the rencofilstat group did, however, grow to the sizes measured in the vehicle, sofosbuvir, and velpatasvir groups. Altogether, the findings demonstrate for the first time that rencofilstat possesses a unique property among anti-HCV agents, i.e., it exerts additional anti-HCC activity independently of its antiviral activity.

## 4. Discussion

In the U.S., HCC represents the fastest-growing cause of cancer mortality [53,54,55,56,57]. HCC accounts for 800,000 deaths per year. Over the past 20 years, the incidence of HCC has more than doubled. Mortality mirrors HCC incidence. An increasing number of young patients have been affected, as the demographic shifts from those with primarily alcoholic liver disease to those in the fifth to sixth decades of life as the consequences of viral hepatitis acquired earlier in life and in conjunction with high-risk behavior [53,54,55,56,57]. In the U.S., the risk factors have historically included alcoholic cirrhosis and viral hepatitis infection. However, the obesity epidemic has resulted in a growing population of patients with nonalcoholic fatty liver disease (NAFLD) and nonalcoholic steatohepatitis (NASH). NAFLD and NASH progress to liver fibrosis, cirrhosis, and HCC. These patients are expected to continuously drive the HCC epidemic worldwide, reflecting the reservoir of the viral hepatitis endemic in the population. Recent studies raised a red flag regarding unexpectedly higher rates of HCC recurrence following treatment with DAAs for HCV infection [58,59,60,61,62,63,64,65,66,67,68,69,70,71]. Studies describing the use of DAAs in HCV patients with HCC are extremely scarce. A recent study assessed the efficacy of DAA regimens in HCV cirrhotic patients who have had HCC compared with those without HCC. Remarkably, 12 weeks post-DAA treatment, patients with liver cancer were eight times more likely to fail DAA treatment than patients without HCC [72]. The authors postulated that HCC serves as a sanctuary for HCV, where viral particles evade DAA therapy [73,74]. The increased risk of HCC development following HCV cure is likely the consequence of changes in inflammation [75,76,77]. For example, DAAs rapidly reduce inflammation but increase serum VEGF levels—a rationale for tumor risk during anti-HCV treatment. Therefore, the identification of drugs that would interfere with the development of viral hepatitis-induced liver damage, especially HCC, would represent critical tools to enhance the efficacy of DAA treatments. Moreover, the underlying mechanisms of viral reactivation during DAA therapy for HCV are poorly understood. There is thus an urgent need for the identification of new drugs with prolonged effectiveness and with distinct MoA that prevent the development of HCV-induced HCC.

This study describes several novel findings. First, it comprehensively describes the development of HCC induced by HCV infection, not by NASH (patients) [15] or by a chemical/high-fat diet-mediated HCC induction (mice) [14]. Second, it shows that four HCV genotypes (GT1 to GT4) induce HCC and that their viral replications remain stable over a period of 30 weeks. Third, it describes the progressive development of HCV-induced HCC over a period of 24 weeks. Fourth, this study investigates the time frame of rencofilstat administration for anti-HCC efficacy and demonstrates that even 16 weeks after HCV infection, rencofilstat still prevents HCV-induced HCC development independently of its antiviral activity. Fifth, this study demonstrates that the cyclophilin inhibitor rencofilstat, but not other potent anti-HCV agents such as NS5A inhibitors or NS5B inhibitors, prevents HCC development 16 h post-HCV infection.

The antiviral MoA of CypIs toward HCV is relatively well understood. Specifically, we and others obtained several lines of evidence suggesting that CypIs inhibit HCV infection by preventing CypA-HCV NS5A interactions, leading to the prevention of the formation of double-membrane vesicles (DMVs), which normally shelter the replication of the viral genome [28]. In sharp contrast, the anti-HCC MoA of CypIs is poorly understood. In the present study, we found that rencofilstat partially inhibited HCV-induced HCC and possibly mostly through mechanisms independent of its antiviral activity, suggesting that the CypI rencofilstat possesses anticancer properties. Supporting this notion, we have previously reported that rencofilstat inhibits nonviral-induced liver damage, including liver steatosis, fibrosis, and HCC, as demonstrated by the prevention of cancerous liver nodule formation [14]. This is in accordance with a recent study providing evidence that rencofilstat exhibits a pronounced antifibrotic effect in a nonviral cohort with metabolic disease [15]. Several scenarios may explain the beneficial anti-HCC effect of rencofilstat. Since Cyps represent the main targets of CypIs, the neutralization of the peptidyl–prolyl isomerase activity of specific members of the Cyp family is likely to be the cause for the anti-HCC activity of rencofilstat. To elucidate which Cyp members participate in HCC development, we developed CypA, CypB, and CypD knockout (KO) mice and recently obtained strong evidence that CypD is the main key player in the development of nonviral-induced HCC (manuscript in preparation). Previous work demonstrated that CypD, which resides within mitochondria, may either enhance or decrease tumor growth [78]. Since CypD enhances aerobic glycolysis by recruiting hexokinase II to the mitochondrial outer membrane [79,80], it may play a key role in cancer cell metabolism. CypD by inhibiting oxidative stress-induced necrosis may inhibit cell death, thus enhancing cancer cell survival [81]. On the other hand, CypD promotes cancer cell death by enhancing mitochondrial permeability transition pore (mPTP)-mediated apoptosis and necrosis [82,83,84]. These conflicting CypD effects on tumor growth likely result from the broad spectrum of its putative binding ligands that may interfere with mitochondrial permeability. Further studies are required to understand how CypD regulates HCC development at molecular and cellular levels. Similarly, additional analyses will be conducted aiming to provide the mechanisms of action for the beneficial effect of rencofilstat on HCV-induced HCC development. This includes liver macrophage differentiation, stellate cell activation, liver inflammation, and RNA-seq of HCC and nontumor liver tissue analyses. These upcoming analyses may shed some light on the anti-HCC mechanisms of action of rencofilstat.

## Figures and Tables

**Figure 1 viruses-15-02099-f001:**
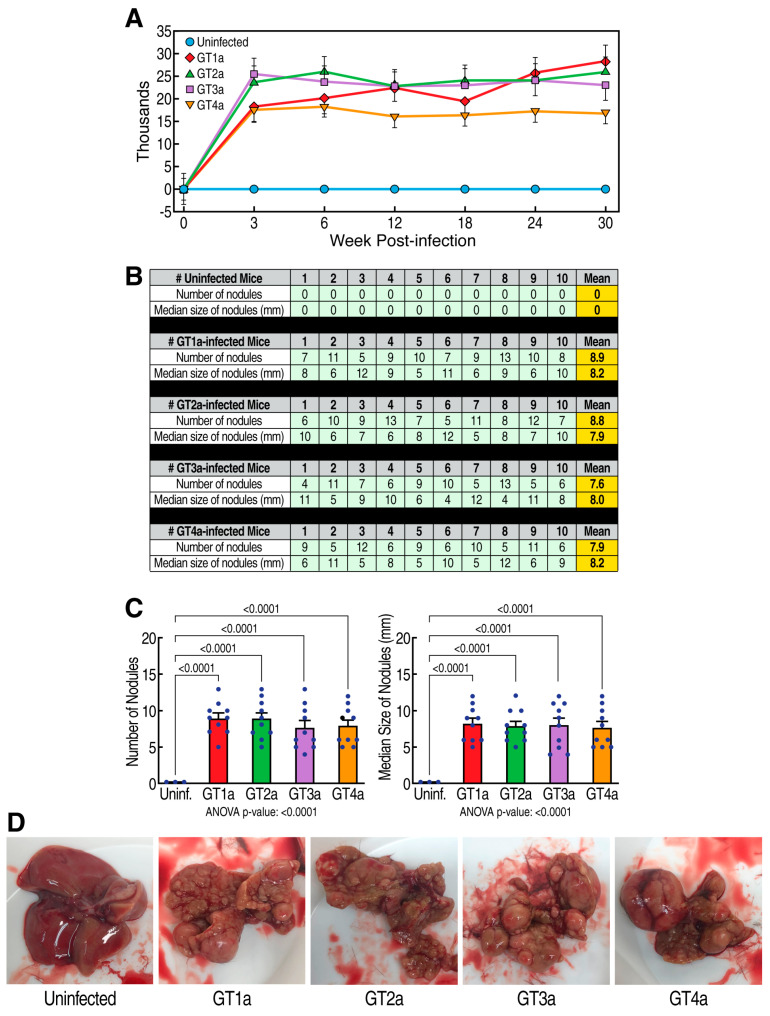
HCV induces HCC: (**A**) Humanized MUP-uPA-SCID/Beige mice (n = 10) were infected with HCV GT1a, GT2a, GT3a, and GT4a, and viral infection was monitored via qRT-PCR at the indicated time points. Data are expressed as HCV RNA copies/mL of serum. (**B**) Nodules were analyzed in fresh livers collected at sacrifice 30 weeks post-HCV infection. Liver nodules were counted, their diameters measured with a ruler and classified as small (0.1 cm in diameter but not exceeding 0.5 cm), medium (0.5–1 cm in diameter), or large (>1 cm in diameter). The cancerous status of each nodule was verified via qRT-PCR for tumor markers MCP-1, Timp-1, and glypican-3 in nodule tissue lysates. (**C**) Statistical analyses of (**B**). (**D**) Representative liver pictures of uninfected and HCV-infected mice.

**Figure 2 viruses-15-02099-f002:**
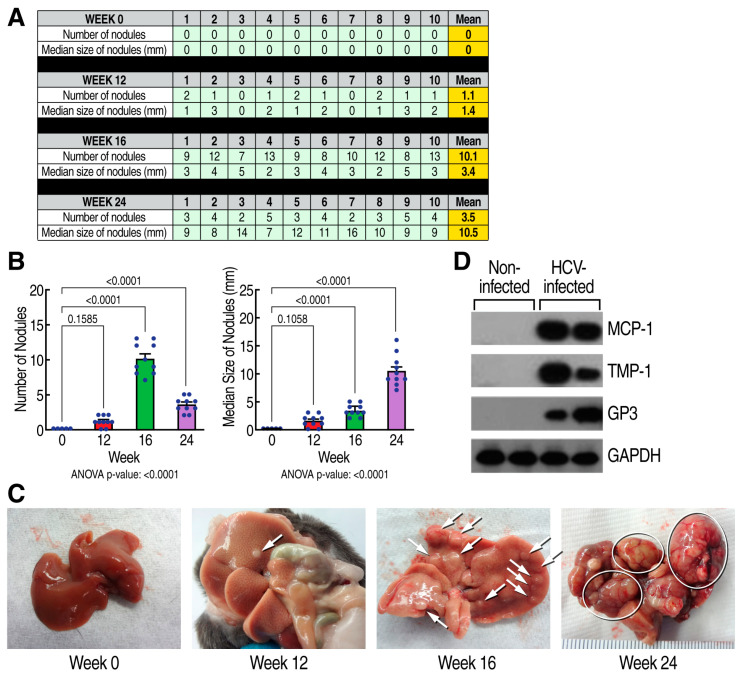
Kinetic of the sequential progression of HCV-induced HCC: (**A**) Humanized MUP-uPA-SCID/Beige mice (n = 10) were infected with HCV GT1a and liver nodules were analyzed at week 0, 12, 16, and 24 post-HCV infection. Liver nodules were counted, their diameters measured, and their cancerous status was verified via qRT-PCR for tumor markers MCP-1, Timp-1, and glypican-3. (**B**) Statistical analysis was conducted. (**C**) Representative liver pictures of uninfected and HCV GT1a-infected mice. (**D**) Western blot analysis of the expression of cancer markers—MCP-1, TIMP-1, and GP3—in either isolated nodules from HCV-infected mice (two left lanes corresponding to two distinct mice) or from liver tissue from uninfected mice (two right lanes corresponding to two distinct mice).

**Figure 3 viruses-15-02099-f003:**
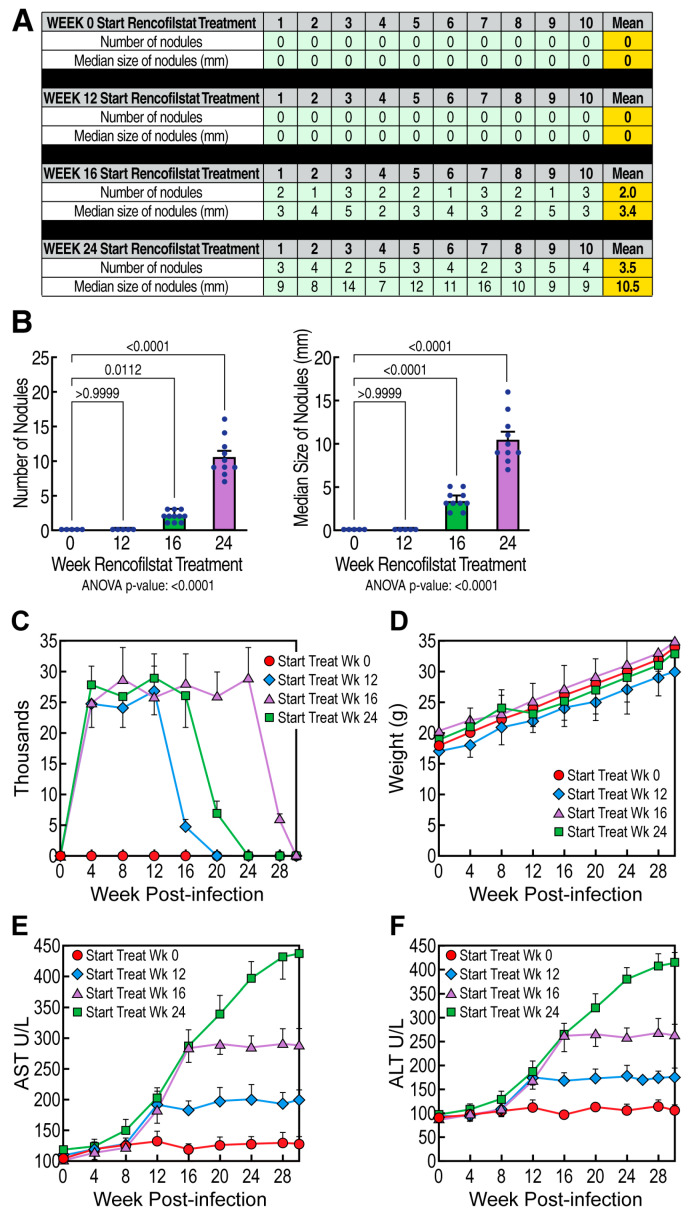
Antitumor activity of rencofilstat in HCV-induced HCC: (**A**) Humanized MUP-uPA-SCID/Beige mice (n = 10) were infected with HCV GT1a and daily treated with rencofilstat starting at week 0, 12, 16, and 24 post-infection. Liver nodules were analyzed at week 24 post-HCV infection. Liver nodules were counted, their diameters were measured and the cancerous status of each nodule was verified via qRT-PCR for tumor markers MCP-1, Timp-1, and glypican-3. (**B**) Statistical analysis was conducted. (**C**) HCV GT1 infection was monitored via qRT-PCR at the indicated time points. Data are expressed as HCV RNA copies/mL of serum. (**D**) Each mouse from each group was weighed every 4 days. (**E**,**F**) AST and ALT serum levels (units/liter) were quantified with ELISA at the indicated time points.

**Figure 4 viruses-15-02099-f004:**
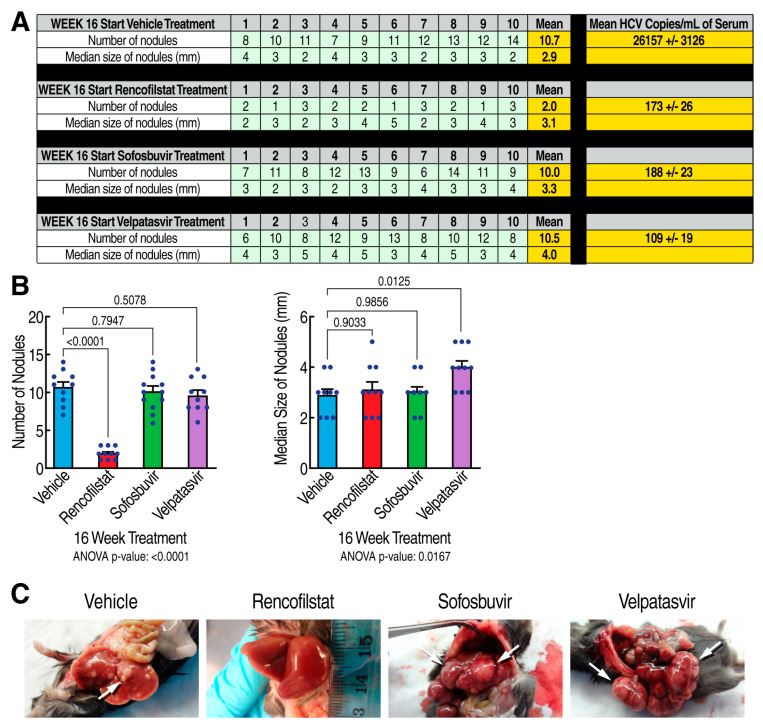
Rencofilstat suppresses HCV-induced HCC mostly independently of its antiviral activity: (**A**) Humanized MUP-uPA-SCID/Beige mice (n = 10) were infected with HCV GT1a and daily treated with the CypI rencofilstat, the NS5Bi sofosbuvir, and the NS5Ai velpatasvir starting at week 16 post-infection. Serum viral loads and liver nodules were analyzed at week 30 post-HCV infection. Liver nodules were counted, their diameters were measured, and the cancerous status of each nodule was verified via qRT-PCR for tumor markers MCP-1, Timp-1, and glypican-3. (**B**) Statistical analysis was conducted. (**C**) Representative liver pictures of uninfected and HCV GT1a-infected mice.

## Data Availability

All data are available within the manuscript.

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
