# Peer review of "The Cyclophilin Inhibitor Rencofilstat Decreases HCV-Induced Hepatocellular Carcinoma Independently of Its Antiviral Activity"

_viruses, 2023, doi:10.3390/v15102099_

Round 1

Reviewer 1 Report

Stauffer and colleagues have studied the effect of the Cyclophilin inhibitor rencofilstat (analog of cyclosporin) on HCV infection and most importantly, HCC development. The authors show in humanized mice that rencofilstat inhibits HCV infection as measured by the serum HCV RNA but more importantly, it decreases the HCC nodule formation which seems to be independent on its antiviral effect. Thus, the authors shed light on the anti-HCC properties of Cyp inhibitors (in the present study rencofilstat). The study is well conducted, the results clearly presented and the manuscript is scientifically sound. Moreover, the authors have an important experience in the domain of Cyps and Cyp inhibitors and the present study is within that framework.

Author Response

please find in the attachment

Reviewer 2 Report

The study submitted by Winston Stauffer and co-workers aims to elucidate the impact of the cyclosporin A analog rencofilstat on the development of HCV-associated hepatocellular carcinoma. Using human liver chimeric mice (human hepatocyte-engrafted MUP-uPA-SCID/Beige), the authors found that all four studied HCV genotypes (1-4) efficiently induced liver cancer after 16 weeks of chronic infection. Daily treatment of the HCV-infected animals with rencofilstat efficiently prevented liver cancer formation if administration started before 12 weeks post infection and attenuated liver cancer formation after treatment start 16 post infection. The authors concluded that the observed anti-tumor effect of rencofilstat is partially independent from the drug-induced HCV cure when compared to tumor burden in mice treated with other direct-acting antivirals.

The results of the manuscript are potentially interesting as it may suggest cyclophilin inhibitor rencofilstat as cancer chemo-preventive agent, which acts partially independent from the presence of HCV. This version of the manuscript is publicly available on bioRxiv (Even though potentially interesting the novelty of the findings seems limited in context of the previous report by the authors of the antiviral impact of rencofilstat (CRV431) in HCV-infected chimeric mice (PMID 32764799) and the recently reported antifibrotic effect of rencofilstat in NASH patients without HCV (PMID 36271849). Therefore, more mechanistic data should be provided elucidating how rencofilstat acts on the disease-relevant non-parenchymal cells (i.e., stellate cells and macrophages activation) and by such potentially attenuating liver fibrosis, steatosis and HCC formation in their animal model (see specific comments).

Specific comments:

(Major)

1.       The viral load kinetics in HCV-infected mice treated with rencofilstat must be provided over the entire 24 weeks.

2.       Potential liver toxicity of the used dose 50/mg/kg/day over 24 weeks must be controlled (liver enzymes, weight). 

3.       To consolidate the qPCR data, the presence of hepatocellular carcinoma in the livers must be confirmed by immuno-histochemistry.

4.       A pronounced anti-fibrotic effect of the compound had been previously reported (PMID36271849) in a non-viral cohort with metabolic disease. Thus, cancer-risk factors like steatosis and fibrosis must be assessed by liver tissue staining.

5.       The impact of rencofilstat on non-parenchymal cells with potential impact on fibrosis and cancer risk must be monitored (e.g., liver macrophage differentiation, stellate cell activation). 

6.       As rencofilstat impacts liver inflammation, appropriate markers must be monitored during the treatment.

7.       It seems that the cancer nodule number but not the nodule size is impacted by the treatment. This may suggest a role of rencofilstat on stress-induced liver regeneration. This would be consistent with previous reports on the involvement of cyclophilin depletion during liver regeneration (PMID 35834573). Bulk RNA-seq of non-tumor liver tissues could help to determine a potential shift from a more progenitor towards mature hepatocyte phenotype or vice verca, e.g., using gene set enrichment analysis.

(Minor)

1.       Figs 3b and 4b-c labels: rencofilstat not reconfilstat

Author Response

please find in the attachment

Round 2

Reviewer 2 Report

The authors responded to all my points in a sufficient manner. These included the emphasis of the key novelties in the manuscript. As requested, the authors provided the viral loads throughout the rencofilstat treatment. However, the statement of the authors that the molecular mode of action is partially virus independent largely depends on one experiment comparing the anti-tumor effects of rencofilstat and known antivirals sofosbuvir and velpatasvir. To control that the HCV antiviral indeed did their job all viral loads in these animals (endpoint) needs to be provided as well. I apologize that I have not requested this during the last round but I belief that this is an important to support the conclusion.

Author Response

We totally agree with the reviewer that the efficacy of the antiviral should be presented. In fact, they are presented in the last column of Figure 4.